# Iterative Analog–Digital Multi-User Equalizer for Wideband Millimeter Wave Massive MIMO Systems

**DOI:** 10.3390/s20020575

**Published:** 2020-01-20

**Authors:** Roberto Magueta, Daniel Castanheira, Pedro Pedrosa, Adão Silva, Rui Dinis, Atílio Gameiro

**Affiliations:** 1Departamento de Eletrónica, Telecomunicações e Informática (DETI), Instituto de Telecomunicações (IT), University of Aveiro, 3810-193 Aveiro, Portugal; dcastanheira@av.it.pt (D.C.); pmfopedrosa@gmail.com (P.P.); asilva@av.it.pt (A.S.); amg@ua.pt (A.G.); 2Instituto de Telecomunicações (IT), Faculdade de Ciências e Tecnologia, University Nova de Lisboa, 1099-085 Lisboa, Portugal; rdinis@lx.it.pt

**Keywords:** iterative analog–digital equalizer, massive MIMO, millimeter-wave communications, hybrid analog–digital architectures

## Abstract

Most of the previous work on hybrid transmit and receive beamforming focused on narrowband channels. Because the millimeter wave channels are expected to be wideband, it is crucial to propose efficient solutions for frequency-selective channels. In this regard, this paper proposes an iterative analog–digital multi-user equalizer scheme for the uplink of wideband millimeter-wave massive multiple-input-multiple-output (MIMO) systems. By iterative equalizer we mean that both analog and digital parts are updated using as input the estimates obtained at the previous iteration. The proposed iterative analog–digital multi-user equalizer is designed by minimizing the sum of the mean square error of the data estimates over the subcarriers. We assume that the analog part is fixed for all subcarriers while the digital part is computed on a per subcarrier basis. Due to the complexity of the resulting optimization problem, a sequential approach is proposed to compute the analog phase shifters values for each radio frequency (RF) chain. We also derive an accurate, semi-analytical approach for obtaining the bit error rate (BER) of the proposed hybrid system. The proposed solution is compared with other hybrid equalizer schemes, recently designed for wideband millimeter-wave (mmWave) massive MIMO systems. The simulation results show that the performance of the developed analog–digital multi-user equalizer is close to full-digital counterpart and outperforms the previous hybrid approach.

## 1. Introduction

The underutilized millimeter-wave (mmWave) frequency spectrum has been explored for future wideband cellular communication networks because there is an overcrowding of conventional sub-6 GHz bands [1]. Together with mmWave, the use of a large or a massive number of antennas allows higher data rates for future wireless networks [2]. Therefore, mmWave communications and massive MIMO (mMIMO) are considered as two key technologies for future 5G communications [3].

The combination of mmWave with mMIMO is very attractive because, when compared to the current communication systems, it has a smaller wavelength, and more antennas can be compacted in the same volume [4]. This combination offers more degrees of freedom, but it also leads to more correlated channels [5], and thus, new and efficient beamforming techniques and spatial multiplexing for both the transmitter and the receiver sides must be exploited [6]. Furthermore, the power consumption and high cost of analog-to-digital converters (ADCs) and digital-to-analog converters (DACs) mixers or power amplifiers for mmWave make it impractical to have one dedicated radio frequency (RF) chain for each transmit and receive antenna [7]. Therefore, the design of new beamforming techniques, different from those adopted for the sub-6 GHz band, is a necessity [8].

A simple approach is the use of fully analog beamforming techniques using only phase shifters to overcome the hardware limitations, allowing low complexity implementations [3]. However, the performance of only analog beamforming techniques is limited, and it is typically only used for single-stream transmission [9]. To overcome the performance limitations, some hybrid analog–digital architectures were proposed in [9], where part of the signal processing is performed in the analog domain, and a reduced-complexity processing is left to the digital domain.

Concerning the modulation scheme, orthogonal frequency division multiplexing (OFDM) is widely employed in current standards. However, OFDM is subject to high amplitude signal fluctuations due to nonlinear distortions caused by the power amplifiers and which degrade the performance of the system. The single-carrier frequency-division multiple access (SC-FDMA) is an interesting approach to deal with this problem, but the major drawback is the residual interference. It is well known that linear equalizers are not the most efficient way to deal with this issue, and nonlinear/iterative equalizers, namely those based on iterative block decision feedback equalization (IB-DFE) principle, have been shown to present better performance [10]. However, the solutions proposed for convectional systems—i.e., for fully digital architectures—cannot be directly used in hybrid architectures where the processing is distributed by the analog and digital domains. Consequently, the design of new iterative analog–digital beamforming schemes for hybrid SC-FDMA based systems is of paramount importance. Therefore, in this paper we adopt a hybrid analog–digital architecture, to achieve a good tradeoff between performance and complexity and we design a fully iterative analog–digital multiuser equalizer for wideband mmWave mMIMO SC-FDMA systems.

### 1.1. Related Works

In this section, we briefly review the state-of-the-art on hybrid analog–digital architectures. Namely, beamforming approaches designed for narrowband single-user mmWave communications systems have been proposed in [11,12,13]. Particularly, the authors of [11] designed a bird swarm algorithm based on a matrix-inversion bypass precoder algorithm to overcome the large complexity of orthogonal matching pursuit method, namely the implicit matrix inversion. In [12], a general solution for single user narrowband systems was proposed to convert any existing precoder/combiner designed for the full digital structure into an analog–digital precoder/combiner for the hybrid structure. The nonconvex problem is decoupled into a series of convex subproblems and then solved by a singular value decomposition-based technique to obtain an initial solution. Next, the phase increment of each entry in the RF precoder/combiner from iteration to iteration is restricted. The approach discussed above assumed a fully connected hybrid architecture, where each RF chain is connected to all antennas. However, this architecture may require a large number of connections; thus, one possible solution is the use of the subconnected hybrid architectures presented in [13], where each RF chain is connected only to a subset of antennas.

Transmit and receive beamforming approaches designed for narrowband multi-user mmWave communications systems have also been proposed in [14,15,16,17,18,19,20,21]. The authors of [14] designed an uplink receive beamforming with single antenna user terminals (UTs) that consider the multi-user interference at two stages: analog and digital. In [15], it was proposed four different digital precoding techniques with a hybrid beamformer for multicell systems, to reduce inter-beam interference. An iterative matrix decomposition based on the subspace projection of angles of departure (AoD) of the channel paths to perform the block diagonalization was proposed for downlink mmWave narrowband channel was proposed in [16] to obtain interference free channels for the UTs. The authors of [17] designed an iterative hybrid precoder and combiner algorithm by exploiting the duality of the uplink and downlink multi-user mmWave MIMO channels. A hybrid weighted minimum mean square error (MMSE) precoding and combining scheme was proposed in [18]. In this algorithm, the hybrid precoder and combiners are optimized and updated in an iterative manner to minimize a weighted sum-MSE cost function. The authors of [19] proposed a linear detector for uplink systems, where in order to avoid the direct matrix inverse of the MMSE, which is computationally expensive, they proposed an improved Jacobi iterative algorithm. This algorithm works by accelerating the convergence rate of the signal detection process, and where a whole-correction method is applied to update the iterative process. In [20], it was proposed a precoder for a downlink multiuser system, with partially connected hybrid beamforming architecture, and where the analog part of precoder is composed by both variable phase shifters and constant phase shifters. Additionally, the sum rate was considered as a metric and a greedy algorithm was employed to reduce the complexity of the algorithm, since the exact solution of the combinatorial problem is mathematically intractable. A hybrid iterative block space-time equalizer, based on IB-DFE principles [22], was proposed [21].

The previous works mainly focused on hybrid beamforming approaches for either single or multi-user narrowband systems. However, the design of a hybrid solution for mmWave mMIMO wideband communications is of paramount importance. Hybrid approaches specifically designed for single user can be found in [23,24,25]. Hybrid precoding solutions and codebooks for limited feedback wideband mmWave systems were discussed in [23]. It was assumed that the digital precoding is performed in the frequency domain and can be different for each subcarrier, and the analog precoder is constant over the subcarriers. To flatten the fading channel over a wide band and maximize the system capacity, a signal-to-interference ratio constrained capacity maximization algorithm to design the precoder and the combiner was proposed in [24]. A closed-form solution for fully connected orthogonal frequency division multiplexing (OFDM) hybrid analog/digital precoding for frequency selective mmWave single user systems was developed in [25]. This approach was then extended to the partially connected case, and a novel technique that dynamically constructs the hybrid subarrays knowing the long-term channel characteristics was proposed. Recently, solutions for wideband mmWave multi-user downlink massive MIMO-OFDM systems were proposed in [26], and for uplink in [27,28,29,30,31]. Hybrid precoders for downlink OFDM wideband mMIMO systems, aimed at minimizing the total transmit power of the base station, subject to both the coverage constraint of signaling and data rate requirements of users were proposed in [26]. The authors of [27] designed a joint spatio–radio resource and three hybrid precoding algorithms for systems with limited feedback: (1) a hybrid precoder with user-beam selection to maximize the sum proportional rate; (2) a low complexity suboptimal solution using limited statistical channel state information (CSI) feedback; (3) a k-mean algorithm based on an unsupervised machine learning scheme. In [28], a hybrid linear equalizer for sub-connected hybrid architectures that minimizes the average BER over all the subcarriers was designed. Also for sub-connected architecture, but using a dynamic subarray antennas, it was designed in [29], a two-step hybrid equalizer, where in the first step, the antennas are dynamically mapped to the RF chains and then, in the second step, an iterative digital equalizer is designed. In [30,31], it was also applied the two-step approach, both for full-connected architectures, using in [30] the constant envelope OFDM (CE-OFDM) modulation technique, and in [31], SC-FDMA.

### 1.2. Contributions

The major novelty of this work is the design of a fully iterative hybrid multi-user equalizer, where both analog and digital parts of equalizer are computed iteratively, allowing better performance than the two-step approaches, with fixed analog part, proposed by the authors in [29,30,31]. Both the analog and digital parts are derived by minimizing the sum of the mean square error, which can be shown to be equivalent to minimizing the weighted difference between the hybrid and the fully digital equalizers. For this, we assume that the analog part is constant over all the subcarriers and the digital part is computed on a per subcarrier basis. Due to the complexity of the optimization problem, we propose an approach to sequentially compute the analog phase shifters for each RF chain, i.e., we first compute the analog coefficients for RF chain 1, then 2, and so on. The computational complexity of the proposed fully iterative analog–digital is higher than the two-step approaches [29,30,31], but its performance is clearly better. Moreover, the simulation results show that the proposed scheme achieves a performance close to the fully digital equalizer.

The remainder of this paper is organized as follows: Section 2 describes the system model adopted in this work. Section 3 describes the analog precoders employed at each UT. In Section 4, we design the hybrid iterative analog–digital multi-user space-frequency equalizer. Section 5 describes the main performance results. Finally, the conclusions are presented in Section 6.

### 1.3. Notations

For any matrix A, denoted by boldface capital letters, or for any column vector a, denoted by boldface lowercase letters, tr(.), (.)*, (.)T, and (.)H denote the trace, the conjugate, transpose and Hermitian of a matrix, respectively. The operator sign(a) represents the sign of real number a; it is applied element-wise to matrices and sign(c)=sign(ℜ(c))+jsign(ℑ(c)) if c∈ℂ. The functions ℜ(c) and ℑ(c) represent the real part and imaginary part of c, respectively. {αl}l=1L represents an L length sequence. The function diag(a) gives a diagonal matrix A, where the entries of the diagonal of A are equal to vector a. The function diag(A) gives a vector equal to the entries of diagonal of the matrix A. The vector a=[aq]1≤q≤Q1∈ℂQ1Q2 and the matrix A=[Aq]1≤q≤Q1∈ℂQ1Q2×L are the concatenation of vector aq∈ℂQ2 and matrix Aq∈ℂQ2×L, respectively. The element of row *n* and column *l* of A is denoted by A(n,l). The matrix IN is the identity matrix N×N. Finally, the indices, *t*, k, and u represent the time domain, subcarrier in the frequency domain, and user terminal, respectively.

## 2. System Model Characterization

In this manuscript, we consider an uplink mmWave system with U users sharing the same radio resources and equipped with a single RF chain and $N_{tx}$ transmit antennas, where each user transmits a single data stream per subcarrier. The base station is equipped with $N_{rx}^{RF}$ RF chains and $N_{rx}$ receive antennas, with $U\leq N_{rx}^{RF}\leq N_{rx}$ [14]. The considered uplink system uses SC-FDMA as the access technique, with $N_{c}$ available subcarriers.

We consider a clustered channel discussed in [23], where the delay—d MIMO channel matrix of the *u*th user, $H_{u,d}$, represents the sum of the contribution of $N_{cl}$ clusters, each of which contribute with $N_{ray}$ propagation paths, and may be expressed as
(1)$$H_{u,d}=\sqrt{\frac{N_{rx}N_{tx}}{\rho_{PL}}}\sum_{q=1}^{N_{cl}}\sum_{l=1}^{N_{ray}}(\alpha_{q,l}^{u}p_{rc}(dT_{s}-\tau_{q}^{u}-\tau_{q,l}^{u})a_{rx}\left(\phi_{q,l}^{rx,u},\theta_{q,l}^{rx,u}\right)a_{tx}{\left(\phi_{q,l}^{tx,u},\theta_{q,l}^{tx,u}\right)}^{H}).$$
where $\alpha_{q,l}^{u}$ is the complex path gain of the *l*th ray in the *q*th scattering cluster, and a raised-cosine filter is adopted for the pulse shaping function $p_{rc}\left(.\right)$ for $T_{S}$-spaced signaling as in [23]. The qth cluster has a time delay $\tau_{q}^{u}$ while each ray l from *q*th cluster has a relative time delay $\tau_{q,l}^{u}$. The angles $\phi_{q,l}^{rx,u}$, $\theta_{q,l}^{rx,u}$, $\phi_{q,l}^{tx,u}$, and $\theta_{q,l}^{tx,u}$ are the azimuth and elevation angles of arrival and departure, respectively. For instance, $\phi_{q,l}^{rx,u}$ has a Laplacian distribution, with mean $\phi_{q}^{rx,u}$ uniformly distributed in [0,2π] and variance $\sigma_{\phi_{q}^{rx,u}}^{2}$. The remaining angles have similar distributions. The paths delay is uniformly distributed in $\left[0,DT_{s}\right]$, and the angles follow the random distribution mentioned in [23], such that $E[{\left\|H_{u,d}\right\|}_{F}^{2}]=N_{rx}N_{tx}$. Finally, the vectors $a_{rx,u}$ and $a_{tx,u}$ denote the receive and transmit array vectors, respectively. For a uniform planar array (UPA) in the *yz*-plane with $N_{y}$ and $N_{z}$ elements on the *y* and *z* axes respectively, the array response vector is given by
(2)$$a_{UPA}\left(\phi ,\theta \right)=\frac{1}{\sqrt{N_{y}N_{z}}}{\left[1,\ldots ,e^{j2\pi \frac{\gamma }{\lambda }d\left\{m\sin\left(\phi \right)\sin\left(\theta \right)+n\cos\left(\theta \right)\right\}},\ldots ,e^{j2\pi \frac{\gamma }{\lambda }d\left\{\left(N_{y}-1\right)\sin\left(\phi \right)\sin\left(\theta \right)+\left(N_{z}-1\right)\cos\left(\theta \right)\right\}}\right]}^{T},$$
where λ is the wavelength, γ is the inter-element spacing, $0\leq m<N_{y}$ and $0\leq n<N_{z}$. The uniform linear array (ULA) is obtained, making $N_{y}=1$ or $N_{z}=1$. $\rho_{PL}$ denotes the path-loss between the transmitter and the receiver separated by a distance d, such that [32]
(3)$$\rho_{PL,dB}=\rho_{PL0,dB}+10n_{p}\log_{10}\left(\frac{d}{d_{0}}\right)+S_{\sigma_{s}},$$
where $\rho_{PL,dB}=10\log_{10}\left(\rho_{PL}\right)$ and $d_{0}$ is the reference distance. $\rho_{PL0,dB}$ represents the free-space loss at distance $d_{0}$, $n_{p}$ is the path-loss exponent and $S_{\sigma_{s}}$ is the log-normal shadowing with a standard deviation of $\sigma_{s}(dB)$.

The frequency domain channel Hu,k∈ℂNrx×Ntx of the *u*th user at the subcarrier *k* can be given by,
(4)Hu,k=∑d=0D−1Hu,de−j2πkNcd,

Which can also be expressed as
(5)$$H_{u,k}=A_{rx,u}\Delta_{u,k}A_{tx,u}^{H},$$
where $\Delta_{u,k}$ is a diagonal matrix, with entries (q,l) that correspond to the paths gain of the *l*th ray in the *q*th scattering cluster. $A_{tx,u}=[a_{tx,u}(\phi_{1,1}^{tx,u},\theta_{1,1}^{tx,u}),\ldots ,a_{tx,u}(\phi_{N_{cl},N_{ray}}^{tx,u},\theta_{N_{cl},N_{ray}}^{tx,u})]$ and $A_{rx,u}=[a_{rx,u}(\phi_{1,1}^{rx,u},\theta_{1,1}^{rx,u}),\ldots ,a_{rx,u}(\phi_{N_{cl},N_{ray}}^{rx,u},\theta_{N_{cl},N_{ray}}^{rx,u}))]$ hold the transmit and receive array response vectors of the *u*th user, respectively.

At the *k*th subcarrier, the received signal is given by
(6)yk=∑u=1UHu,kxu,k+nk,
where nk∈ℂNrx denotes the zero mean Gaussian noise, with variance σn2, and xu,k∈ℂNtx represents the discrete transmitted complex baseband signal of the *u*th user at subcarrier k.

## 3. Transmitter Design

In this section, we describe the proposed transmitter; the receiver design will be discussed in the next section. In Figure 1, we present the block diagram of the *u*th user terminal. We consider M-QAM constellations, where the data symbols $s_{u,t}$, have $E[|s_{u,t}{|}^{2}]=\sigma_{u}^{2}.$ The sequence ${\left\{s_{u,t}\right\}}_{t=1}^{N_{c}}$ is divided into R data blocks of size $S=N_{c}/R$, where ${\left\{s_{u,t}\right\}}_{t=(r-1)S+1}^{rS}$ denotes the *r*th data block, and the sequence ${\left\{c_{u,k}\right\}}_{k=(r-1)S+1}^{rS}$ is the DFT of ${\left\{c_{u,t}\right\}}_{t=(r-1)S+1}^{rS}$ a transformation from the time domain to the frequency domain. After the time-frequency transformation the frequency domain data is interleaved and mapped to the OFDM symbols. Finally, the cyclic prefix (CP) is inserted in the blocks associated with each RF chain. Since the proposed hybrid equalizers detect each *S*-length data block independently, let us focus on a single *S*-length data block in order to simplify the math formulation [10]. Therefore, hereinafter the index *r* is not considered and the formulation is done for the frequency domain *S*-length block ${\left\{c_{u,k}\right\}}_{k=1}^{S}$, which is the DFT of the time domain sequence ${\left\{s_{u,t}\right\}}_{t=1}^{S}$.

The analog precoder is modeled mathematically by the vector fa,u∈ℂNtx. Because of hardware constraints, only analog phase shifters are employed. This forces all elements of the vector fa,u to have equal norm (|fa,u(n)|2=Ntx−1) The analog precoder is independent of the subcarrier index *k*, i.e., it is the same for all subcarriers. These assumptions are followed by most of the recently works on hybrid massive MIMO mmWave based systems [23,24,25].

The discrete transmit complex baseband signal xu,k∈ℂNtx of the *u*th user at subcarrier k may be mathematically expressed by
(7)xu,k=fa,ucu,k,
where cu,k∈ℂ.

Since the aim is to design a multi-user equalizer, in this paper, we consider the pure analog precoders discussed in [29,31], which are briefly discussed here. In the first case, presented in [31], we consider that the UT has no access to CSI, which simplifies the overall design, and in the second case, designed in [29], we consider an analog precoder based on the knowledge of partial CSI at the transmitter, i.e., only the average AoD is assumed to be known at the transmitters.

### 3.1. Analog Precoder Design: No CSI at Users Terminals

In this case, the analog precoder vector is generated randomly for the *u*th user according to [31]
(8)$$f_{a,u}={\left[e^{j2\pi \omega_{n}^{u}}\right]}_{1\leq n\leq N_{tx}},$$
where $\omega_{n}^{u}$, $n\in \{1,\ldots ,N_{tx}\}$, u∈{1,…,U} are i.i.d. uniform random variables with support $\omega_{n}^{u}\in [0,1]$.

### 3.2. Analog Precoder Design: Average AoD knowledge at User Terminals

For the second case, it is assumed that the *u*th user has knowledge of the average AoD, ϕqtx,u,θqtx,u,q=1,…,Ncl, of each cluster of its own channel. Based on this knowledge, user *u* computes the correlation matrix A¯tx,uA¯tx,uH, where the matrix A¯tx,u is given by [29]
(9)A¯tx,u=[atx,u(ϕ1tx,u,θ1tx,u),…,atx,u(ϕqtx,u,θqtx,u),…,atx,u(ϕNcltx,u,θNcltx,u)]∈ℂNtx×Ncl,
with atx,u(θqu) computed from (2) for the UPA case. Let A¯tx,uA¯tx,uH=Λtx,uΣtx,uΛtx,uH be the eigenvalue decomposition of the previous correlation matrix, thus the analog precoder vector of the *u*th user is set as
(10)fa,u(n)=1Ntxexp{j arg(Λtx,u(n,1))},
with 1≤n≤Ntx. As mentioned only partial CSI (average AoD’s, ϕqtx,u and θqtx,u) is required at each UT, which may be obtained at the base station, coded with b bits each one, and then, sent to each user terminal by a feedback channel. For instance, for a channel with Ncl=4 and b=4, this results just in a feedback of 2×4×4=32 bits. Notice that the channel correlation-based approaches would require the feedback and/or estimation of the full correlation matrix, which has Ntx2 entries. Then, the proposed analog precoder has a clear advantage compared to the several channel correlation SVD-based beamforming proposals which have been proposed in the literature [25], since it has both low-complexity and low CSI requirements.

## 4. Analog–Digital Receiver Design

In this section, we start by deriving the proposed fully iterative multiuser equalizer, and then, a complexity comparison with the sub-optimal two-step approaches designed for full connected architecture [31] is presented.

### 4.1. Iterative Analog–Digital Equalizer

We assume at the receiver, a hybrid iterative S-length block space-frequency decoder, as shown in Figure 2. At the *k*th subcarrier, the corresponding concatenated received signal of all users, c˜k(i)=[c˜u,k(i)]1≤u≤U∈ℂU, is
(11)c˜k(i)=Wd,k(i)(Wa(i))Hyk−Bd,k(i)c^k(i−1),
where Wa(i)∈ℂNrx×NrxRF denotes the analog part of feed-forward matrix for the *i*th iteration, and Wd,k(i)∈ℂU×NrxRF and Bd,k(i)∈ℂU×U denote the digital part of the feed-forward and feedback matrices, computed at the *i*th iteration at the *k*th subcarrier, respectively. The vector c^k(i)=[c^u,k(i)]1≤u≤U∈ℂU, where the block {c^u,k(i)}k=1S is the FFT of the block of time domain estimates conditioned to the detector output for user *u* and iteration {s^u,t(i)}t=1S
*i*. The vector s^k(i)=[s^u,t(i)]1≤u≤U, where s^u,t(i) is the hard decision associated with the data symbols of user *u* at iteration *i.*

The received signal is first processed through the analog phase shifters, with $|W_{a}(n,l){|}^{2}=N_{rx}^{-1}$, and then follows the baseband processing composed by $N_{rx}^{RF}$ RF chains. The digital baseband processing includes a feedback closed-loop that employs a forward path and a feedback path for each subcarrier. In the forward path, the signal first passes through a linear filter $W_{d,k}^{\left(i\right)}$ used at subcarrier k and then follows SC-FDMA decoding and data demodulation. The data recovered from the forward path are modulated, and SC-FDMA encoded in the feedback path, and then, the data pass through the feedback filter $B_{d,k}^{\left(i\right)}$ employed at subcarrier k.

From the central limit theorem the entries of vector, ck=[cu,k]1≤u≤U∈ℂU, k∈{1,…,S} are Gaussian distributed, then as the input–output relationship between variables ck and c^k(i), is memoryless, thus by the Bussgang theorem [33] is approximately given by
(12)c^k(i)≈Ψ(i)ck+ϵ^k(i), k∈{1,…,S},
where ϵ^k(i) is a zero mean error vector uncorrelated with ck,
k∈{1,…,S}, and Ψ(i)∈ℂU×U is a diagonal matrix whose *u*th element gives a blockwise reliability measure of the estimates of *u*th block, associated to the *i*th iteration [22]. The coefficients of each block can be estimated at the receiver, as discussed in [10].

The error between the estimated signal before the S-IFFT c˜k(i) given by (11) and the transmit signal after the S-FFT
ck may be expressed by
(13)ϵ˜k(i)=c˜k(i)−ck=(Wad,k(i)Hk−IU−Bd,k(i)Ψ(i−1))ck︸Residual ISI−Bd,k(i)ϵ^k(i−1)︸Errors from estimate c^k(i)+Wad,k(i)nk︸Channel Noise,
where Hk=[Hu,kfa,u]1≤u≤U∈ℂNrx×U is the concatenated equivalent channel, and Wad,k(i)=Wd,k(i)(Wa(i))H. From (13), we can identify three error terms: (1) the residual intersymbol interference (ISI); (2) the error from the incorrect estimate made by c^k of the signal ck; and (3) the part that corresponds to the channel noise. From (13), as we can see in the Appendix A, we obtain the corresponding mean square error for the *k*th subcarrier.
(14)MSEk(i)=E[||c˜k(i)−ck||2]=‖Wad,k(i)Hk−Bd,k(i)Ψ(i−1)−IU‖F2σu2+‖Bd,k(i)(IU−|Ψ(i−1)|2)1/2‖F2σu2+‖Wad,k(i)‖F2σn2.

From (14), we can obtain a semi-analytical BER approximation for an M-QAM constellation with Gray mapping, given by [34]
(15)$$\mathrm{BER}=\frac{\alpha }{US}\sum_{u=1}^{U}\sum_{k=1}^{S}Q\left(\sqrt{\beta {\left({\mathrm{MSE}}_{k,u}^{\left(i\right)}\right)}^{-1}}\right),$$
where $\alpha =4(1-1/\sqrt{M})/\log_{2}[M]$, and Q(.) denotes the Q-function. ${\mathrm{MSE}}_{k,u}^{\left(i\right)}$, such that $\sum_{u=1}^{U}{\mathrm{MSE}}_{k,u}^{\left(i\right)}={\mathrm{MSE}}_{k}^{\left(i\right)}$, is the mean square error on samples ${\tilde{c}}_{k,u}^{\left(i\right)}$ at iteration *i.*

The optimization problem can be given by
(16)$$\begin{array}{l} \left(W_{a}^{\left(i\right)},W_{d,k}^{\left(i\right)},B_{d,k}^{\left(i\right)}\right)\\ \;=\arg\;\min\;\sum_{k=1}^{S}{\mathrm{MSE}}_{k}^{\left(i\right)}\\ \;s.t.\;\sum_{k=1}^{S}\mathrm{diag}(W_{d,k}^{\left(i\right)}{\left(W_{a}^{\left(i\right)}\right)}^{H}H_{k})=SI_{U}\\ \;W_{a}^{\left(i\right)}\in W_{a}, \end{array}$$
where $W_{a}$ denotes the set of feasible analog coefficients, i.e., the $N_{rx}\times N_{rx}^{RF}$ matrices with constant-magnitude entries. The amplitude constraint in (16) is justified by the fact that, if we only consider the MSE minimization, then it may lead to biased estimates [10]. Note that the optimization of (16) considers as a metric the average MSE of the S subcarriers. Because the feedback matrix $B_{d,k}^{\left(i\right)}$ is independent of the constraints of the optimization problem (16), the digital feedback matrix can be designed by minimizing the ${\mathrm{MSE}}_{k}^{\left(i\right)}$ as
(17)$$B_{d,k}^{\left(i\right)}=\arg\;\min\;\sum_{k=1}^{S}{\mathrm{MSE}}_{k}^{\left(i\right)}.$$

From the KKT conditions of the previous problem i.e., $\partial \left(\sum_{k=1}^{S}{\mathrm{MSE}}_{k}^{\left(i\right)}\right)/\partial \left(B_{d,k}^{\left(i\right)}\right)=0$, the solution is
(18)$$B_{d,k}^{\left(i\right)}=\left(W_{d,k}^{\left(i\right)}{\left(W_{a}^{\left(i\right)}\right)}^{H}H_{k}-I_{U}\right){\left(\Psi^{\left(i-1\right)}\right)}^{H}.$$

Replacing (18) in (14), the ${\mathrm{MSE}}_{k}^{\left(i\right)}$ is up to a constant equal to [29]
(19)$${\bar{\mathrm{MSE}}}_{k}^{\left(i\right)}={\left\|\left(W_{d,k}^{\left(i\right)}{\left(W_{a}^{\left(i\right)}\right)}^{H}-{\bar{W}}_{fd,k}^{\left(i\right)}\right){\left({\tilde{R}}_{k}^{\left(i-1\right)}\right)}^{1/2}\right\|}_{F}^{2}.$$
(20)$${\bar{W}}_{fd,k}^{\left(i\right)}=(I_{U}-|\Psi^{\left(i-1\right)}{|}^{2})H_{k}^{H}{\left({\tilde{R}}_{k}^{\left(i-1\right)}\right)}^{-1},$$
(21)$${\tilde{R}}_{k}^{\left(i-1\right)}=H_{k}(I_{U}-|\Psi^{\left(i-1\right)}{|}^{2})H_{k}^{H}+\sigma_{n}^{2}\sigma_{u}^{-2}I_{N_{rx}},$$
where ${\bar{W}}_{fd,k}^{\left(i\right)}$ denotes the non-normalized full-digital equalizer [29]. Therefore, the analog and digital parts of the feed-forward matrix are the solution of the following optimization problem
(22)$$\begin{array}{l} \left(W_{a}^{\left(i\right)},W_{d,k}^{\left(i\right)}\right)=\\ \;\arg\;\min\;\sum_{k=1}^{S}{\bar{\mathrm{MSE}}}_{k}^{\left(i\right)}\\ \;s.t.\;\sum_{k=1}^{S}\mathrm{diag}(W_{d,k}^{\left(i\right)}{\left(W_{a}^{\left(i\right)}\right)}^{H}H_{k})=SI_{U}.\\ \;W_{a}^{\left(i\right)}\in W_{a}\;. \end{array}$$

As optimization problem (22) is nonconvex, an optimum solution is difficult to obtain. Nevertheless, in the following section, we propose a method to obtain an approximate solution to this optimization problem.

#### 4.1.1. Digital Feed-Forward Equalizer Design

First, we compute the feed-forward digital part of the equalizer as a function of the analog matrix, $W_{a}^{\left(i\right)}$. According to (22), for a given analog equalizer matrix $W_{a}^{\left(i\right)}$, the optimum digital part of the equalizer for the *k*th subcarrier is the solution of the following convex optimization problem
(23)$$\begin{array}{l} W_{d,k}^{\left(i\right)}[W_{a}^{\left(i\right)}]=\\ \;\arg\;\min\;\sum_{k=1}^{S}{\bar{\mathrm{MSE}}}_{k}^{\left(i\right)}\\ \;s.t.\;\sum_{k=1}^{S}\mathrm{diag}(W_{d,k}^{\left(i\right)}{\left(W_{a}^{\left(i\right)}\right)}^{H}H_{k})=SI_{U}, \end{array}$$
whose solution is [29]
(24)$$W_{d,k}^{\left(i\right)}[W_{a}^{\left(i\right)}]=\Omega_{d}H_{k}^{H}W_{a}^{\left(i\right)}{\left(R_{d,k}^{\left(i-1\right)}\right)}^{-1},$$
with
(25)$$\Omega_{d}=S{\left(\sum_{k=1}^{S}\mathrm{diag}\left(H_{k}^{H}W_{a}^{\left(i\right)}{\left(R_{d,k}^{\left(i-1\right)}\right)}^{-1}{\left(W_{a}^{\left(i\right)}\right)}^{H}H_{k}\right)\right)}^{-1},$$
(26)$$R_{d,k}^{\left(i-1\right)}={\left(W_{a}^{\left(i\right)}\right)}^{H}{\tilde{R}}_{k}^{\left(i-1\right)}W_{a}^{\left(i\right)},$$
where $\Omega_{d}$ is used to normalize the received power.

#### 4.1.2. Analog Feed-Forward Equalizer Design

To optimize the analog part of the equalizer, we consider an iterative procedure that at step *r* selects the column *r* of matrix Wa(i) from the dictionary Arx∈ℂNrx×NclNrayU given by Arx=[Arx,1,…,Arx,U], with Arx,u defined in Section 2. Please note that on each iteration *i*, the matrix Wa(i) is computed sequentially in *r* steps, one for each RF chain, i.e., we first compute the analog coefficients for RF chain 1, then 2, and so on.

Let Wd,k,r(i)=[wd,k,1(i),…,wd,k,r(i)]∈ℂU×r, Wa,r(i)=[wa,1(i),…,wa,r(i)]∈ℂNrx×r and Wad,k,r(i)=Wd,k,r(i)(Wa,r(i))H where wd,k,r(i)∈ℂU and wa,r(i)∈ℂNrx. Since Wd,k,r(i)=[Wd,k,r−1(i),wd,k,r(i)] and Wa,r(i)=[Wa,r−1(i),wa,r(i)], it follows that
(27)Wad,k,r(i)=Wad,k,r−1(i)+wd,k,r(i)(wa,r(i))H,
for r=1,…,NrxRF. At step *r* of the algorithm Wad,k,r−1(i)=Wd,k,r−1(i)(Wa,r−1(i))H is known because the r−1 columns of matrix Wa,r−1(i) were already selected in the previous steps, and Wd,k,r−1(i) is set to its optimum value accordingly to optimization problem (22), i.e.,Wd,k,r−1(i)=Wd,k(i)[Wa,r−1(i)], see (24). Replacing (27) in (19), the problem (22) simplifies to
(28)wa,r(i)= arg min ∑k=1S‖(wd,k,r(i)(wa,r(i))H−Wres,k,r−1(i))(R˜k(i−1))1/2‖F2,  s.t. wa,r(i)∈ℱa
where ℱa represents the dictionary defined by columns of Arx and Wres,k,r−1(i)=W¯fd,k(i)−Wad,k,r−1(i) is the residue matrix. Note that the normalization amplitude constraint in (22) was removed in (28) because it was already taken into account in the derivation of the digital part of the equalizer (see (24) and (25)). As described in Appendix B, the optimization problem (28) is equivalent to
(29)wa,r(i)=arg max ∑k=1S‖Wres,k,r−1(i)R˜k(i−1)wa,r(i)‖2‖(R˜k(i−1))1/2wa,r(i)‖2, s.t. wa,r(i)∈ℱa.

Therefore, $w_{a,r}^{\left(i\right)}$ is selected as the element of the codebook that maximizes the previous metric. As the codebook ${ℱ}_{a}$ elements are the columns of matrix $A_{rx}$ the index of the best element, denoted by $n_{opt,r},$ may be extracted as
(30)$$n_{opt,r}=\arg\max_{l=1,\ldots ,N_{CB}}\sum_{k=1}^{S}\frac{{\left[\Pi_{k,r}^{H}\Pi_{k,r}\right]}_{l,l}}{{\left[\Gamma_{k,r}^{H}\Gamma_{k,r}\right]}_{l,l}^{}},$$
where $\Pi_{k,r}={\bar{W}}_{res,k,r-1}^{\left(i\right)}A_{rx}$, $\Gamma_{k,r}={\left({\tilde{R}}_{k}^{\left(i-1\right)}\right)}^{1/2}A_{rx}$ and ${\bar{W}}_{res,k,r}^{\left(i\right)}=W_{res,k,r}^{\left(i\right)}{\tilde{R}}_{k}^{\left(i-1\right)}$. Note that the number of paths in mmWave channels is usually small, i.e., the complexity of this selection process is small because there only are $N_{CB}=N_{cl}N_{ray}U$ possible vectors in the dictionary, among which will be selected the $N_{rx}^{RF}$ vectors to build $W_{a,N_{rx}^{RF}}^{\left(i\right)}={\left[w_{a,r}^{\left(i\right)}\right]}_{1\leq r\leq N_{rx}^{RF}}$.

The pseudo-code for the proposed hybrid fully iterative receiver is presented in Algorithm 1, where initially it is assumed $\Psi^{\left(0\right)}=0_{U}$, since in the first iteration we do not have any estimates. For $\Psi^{\left(0\right)}=0_{U}$, the iterative digital part of the equalizer reduces to the standard MMSE equalizer. The algorithm can be summarized as follows. We have an outer loop with $i_{max}$ iterations and an inner loop with $N_{rx}^{RF}$ steps. Let us consider iteration i of the outer loop. For this iteration the non-normalized full-digital equalizer for iteration *i* is computed and the residue matrix is set to the trivial value ${\bar{W}}_{res,k,0}^{\left(i\right)}={\bar{W}}_{fd,k}^{\left(i\right)}{\tilde{R}}_{k}^{\left(i-1\right)}$, i.e., $W_{ad,k,0}^{\left(i\right)}=0$. Next, in the inner loop, we select the best column from the dictionary for the RF chain *r* to build the analog part of feedforward hybrid equalizer and compute the digital part according to (24). Then, the residue matrix is updated and the previous steps are repeated for $r=1,\ldots ,N_{rx}^{RF}$, i.e., until the full analog and digital parts of the feedforward hybrid equalizer matrix are obtained. After that, we compute the digital feedback matrix, and estimate the transmitted data symbols. Finally, we update ${\hat{c}}_{k}^{\left(i\right)}$ and $\Psi^{\left(i\right)}$, and all these step are repeated until the maximum number of iterations is reached in the outer loop.
**Algorithm 1** The proposed fully iterative hybrid space-frequency multi-user algorithm for broadband mmWave mMIMO systems**1:****for**$i=1,\ldots ,i_{max}$**do****2:**
**Compute**${\bar{W}}_{fd,k}^{\left(i\right)}$ accordingly to (20)**3:**
${\bar{W}}_{res,k,0}^{\left(i\right)}={\bar{W}}_{fd,k}^{\left(i\right)}{\tilde{R}}_{k}^{\left(i-1\right)}$**4:**
$W_{a,0}^{\left(i\right)}$**= Empty Matrix****5:**
**for**$r=1,\ldots ,N_{rx}^{RF}$**do****6:**

$\Pi_{k,r}={\bar{W}}_{res,k,r-1}^{\left(i\right)}A_{rx}$; $\Gamma_{k,r}={\left({\tilde{R}}_{k}^{\left(i-1\right)}\right)}^{1/2}A_{rx}$**7:**

$n_{opt,r}=\arg\max_{l=1,\ldots ,N_{CB}}\sum_{k=1}^{S}\frac{{\left[\Pi_{k,r}^{H}\Pi_{k,r}\right]}_{l,l}}{{\left[\Gamma_{k,r}^{H}\Gamma_{k,r}\right]}_{l,l}^{}}$**8:**

$W_{a,r}^{\left(i\right)}=[W_{a,r-1}^{\left(i\right)}|A_{rx}^{\;(n_{opt,r})}]$**9:**

$W_{a,r}^{\left(i\right)}=[W_{a,r-1}^{\left(i\right)}|A_{rx}^{\;(n_{opt,r})}]$**10:**

$W_{d,k,r}^{\left(i\right)}=W_{d,k}^{\left(i\right)}[W_{a,r}^{\left(i\right)}]$**11:**

${\bar{W}}_{res,k,r}^{\left(i\right)}=({\bar{W}}_{fd,k}^{\left(i\right)}-W_{ad,k,r}^{\left(i\right)}){\tilde{R}}_{k}^{\left(i-1\right)}$**12:**
**end for****13:**
$B_{d,k}^{\left(i\right)}=\left(W_{d,k}^{\left(i\right)}{\left(W_{a}^{\left(i\right)}\right)}^{H}H_{k}-I_{U}\right){\left(\Psi^{\left(i-1\right)}\right)}^{H}$**14:**
${\tilde{c}}_{k}^{\left(i\right)}=W_{d,k}^{\left(i\right)}{\left(W_{a}^{\left(i\right)}\right)}^{H}y_{k}-B_{d,k}^{\left(i\right)}{\hat{c}}_{k}^{\left(i-1\right)}$**15:**
**Compute**${\hat{c}}_{k}^{\left(i\right)}$**and**$\Psi^{\left(i\right)}$**16:****end for**

### 4.2. Complexity Comparison

In this section, we compare the complexity of the proposed algorithm with the two-step approach proposed in [31]. This analysis may be divided in two parts, one for the computation of the analog, and the other for the digital component of the equalizer. For Algorithm 1 both the digital and analog components are computed in a given iteration *i*. On the other hand, for two-step only the digital component is updated in iteration *i*. The analog part is only computed once. The update of the digital component needs the inversion of a U×U matrix, and therefore has complexity $O(U^{3})$. The computation of the analog component requires the evaluation of the metric in (27) for all elements of the codebook ${ℱ}_{a}$. The complexity of a matrix-vector product of sizes $U\times N_{rx}$ and $N_{rx}$, respectively, is $O(N_{rx}U)$. As codebook ${ℱ}_{a}$ has $N_{CB}=N_{cl}N_{ray}U$ elements, then the complexity of the metric evaluation is $O(N_{CB}N_{rx}U).$ As previously mentioned both the analog and digital components are updated in each iteration of Algorithm 1, which means that its computational complexity is $O(i_{max}(U^{3}+N_{CB}N_{rx}U))$. Note that $i_{\max}$ denotes the maximum number of iterations. For two-step only the digital part is updated in all $i_{\max}$ iterations, then the computational complexity of this algorithm is $O(i_{max}U^{3}+N_{CB}N_{rx}U)$. As the number of receive antennas is larger than the number of user and $N_{CB}=N_{cl}N_{ray}U\geq U$ follows that $N_{CB}N_{rx}U\geq U^{3}$. Therefore, the complexity of Algorithm 1 and two-step simplify to $O(i_{max}N_{CB}N_{rx}U)$ and $O(N_{CB}N_{rx}U)$, respectively. Hence, the two-step is approximately $i_{\max}$ times faster than Algorithm 1. However as shown in the next section, the proposed fully iterative analog–digital equalizer clearly outperforms the two-step approach.

Additionally, in the next section, we also made a performance comparison with the algorithm of [14], extended here to broadband SC-FDMA systems. In the Gram–Schmidt orthogonalization is done $N_{rx}U^{2}$ multiplications and U(U−1)/2 divisions, therefore, we have $O(N_{rx}U^{2}+U(U-1)/2)$. Then, the analog equalizer matrix is computed by $N_{rx}U$ multiplications, $2N_{rx}U$ divisions and $N_{rx}U$ square roots, which means that $O(4N_{rx}U)$. Finally, in digital part of the equalizer it is used the linear MMSE, which is equivalent to iteration 1 of Algorithm 1, and then $O(U^{3})$. The total complexity is $O(U^{3}+(N_{rx}+0.5)U^{2}+(4N_{rx}-0.5)U)$, where for $N_{rx}\gg 0.5$, it is approximately $O((U^{2}N_{rx}^{-1}+U+4)N_{rx}U)$. Hence, the algorithm of [14] is approximately $i_{\max}N_{cl}N_{ray}U/(U^{2}N_{rx}^{-1}+U+4)$ times faster than Algorithm 1.

## 5. Performance Results

In this section, we show the BER performance of the proposed receiver structure for both analog precoders designed in Section 2, whose parameters are presented in Table 1. The results presented from Figure 3, Figure 4, Figure 5, Figure 6, Figure 7, Figure 8, Figure 9 and Figure 10 do not consider either path loss or shadowing effects between the UTs and the base station. The path loss and shadowing effects are evaluated in Figure 11 and Figure 12, for $n_{p}=4.17$ and $\sigma_{s}=9\;\mathrm{dB}$ [32]. We consider the wideband mmWave channel model defined in (1), perfect synchronization, and CSI at the receiver side. To evaluate the proposed hybrid multi-user equalizers, we consider the analog precoders discussed in Section 2. The analog precoder is generated either randomly according to (8), referred to here as the Non-CSI precoder (NCSI precoder) or the one computed on the basis of the average AoD of each cluster according to (10), referred to as the Partial CSI precoder (PCSI precoder). The performance results are presented in terms of the average BER as a function of $E_{b}/N_{0}$, where $E_{b}$ denotes the average bit energy, and $N_{0}$ denotes the one-sided noise power spectral density. It was assumed that $\sigma_{1}^{2}=\ldots =\sigma_{U}^{2}=1$ and that the average $E_{b}/N_{0}$ is identical for all users u∈{1,…,U} and is given by $E_{b}/N_{0}=\sigma_{u}^{2}/(2\sigma_{n}^{2})=\sigma_{n}^{-2}/2$.

First, let us analyze the results of the proposed fully iterative analog–digital equalizer (Algorithm 1 in the legends) for the referred two precoders, with no path loss or shadowing effects between the UTs and the base station. The curve for fully digital equalizer discussed in [29] is added because it can be considered as a lower bound for the hybrid architectures. The results are also compared with the semi-analytical BER approximation (15). We only added the theoretical curves for the first and fourth iterations for clarity. From the Figure 3 and Figure 4 we can see that the theoretical curves almost overlap with the simulated ones, which means that our semi-analytical approach is quite accurate. As also observed in Figure 3 and Figure 4, the performance improves for both cases as the number of iterations increases. We may also see from these two figures that the performance gap from iteration 1 to iteration 2 is higher than from iteration 3 to iteration 4. This change occurs because most of the residual multi-user and intersymbol interferences are mitigated from the first to the second iteration. From the third to the fourth iteration, there is also a benefit from residual interference removal; however, the gain is smaller because most of the interference was previously removed. We can also verify that the gap between NCSI (Figure 3) and PCSI (Figure 4) precoders at a target BER of is 5.3 dB for the fourth iteration, i.e., with only the knowledge of average AoD information at the UTs with the proposed efficient hybrid equalizers, the average BER performance has a significant improvement. Let us compare the average BER performance between the proposed Algorithm 1 and the fully digital iterative multi-user equalizer. Figure 3 and Figure 4 show that, for iteration 4, the proposed Algorithm 1 almost achieves the fully digital bound, i.e., the proposed receive and transmit structures are sufficiently efficient to overtake the constraints imposed by the hybrid architectures.

In Figure 5 and Figure 6, we compare the performance of the proposed iterative hybrid multi-user equalizer, against the performance of two-step approach proposed in [31], and the scheme proposed in [14], where the analog part is computed by applying the Gram–Schmidt (GS) method, while in the digital part a MMSE equalizer is used, referred here as GS/MMSE approach. Starting by analyzing the first iteration, we can see that the BER performance of the two-step approach and Algorithm 1 are similar, since for iteration 1, both algorithms assume $\Psi^{\left(0\right)}=0_{U}$, and then they are equivalent. Focusing now on the fourth iteration, the performance of two-step approach is far from the performance of Algorithm 1. This happens because only the digital part is iteratively, while the proposed algorithm the analog and digital parts are both computed in each iteration, and thus it is more efficient to remove the interferences. From the figure, we can also observe that the performance of the GS/MMSE approach is approximately the same as the one obtained by our schemes for the first iteration.

This can be explained by the fact that in our schemes the digital part, for the first iteration, falls in the MMSE equalizer.

Now, let us analyze the results presented in Figure 7 and Figure 8 using 16-QAM and considering PCSI precoder. Figure 7 presents the results for Algorithm 1 for 1 up to 6 iterations. As for the QPSK, the performance improves, as the number of iterations increases. Comparing these results with the ones obtained in Figure 4 for QPSK we observe a performance penalty since the 16-QAM is more prone to errors and therefore more iterations are required to remove the residual multi-user and intersymbol interferences. However, similarly to the QPSK case, Algorithm 1 almost achieves the fully digital bound for 16-QAM, requiring only a few iterations. Figure 8 presents the results for both Algorithm 1, the two-step approach considering first and fourth iterations and the GS/MMSE for the first iteration. Comparing these results with the ones presented in Figure 6 for QPSK the same conclusions can be reached.

In Figure 9 and Figure 10 we present the results for the same schemes of the Figure 3 and Figure 4 but now considering UPA configuration. From them, we can see that the theoretical curves almost overlap with the simulated ones, which means that our semi-analytical approach is also quite accurate for the UPA configuration. Moreover, the performance of Algorithm 1 and the fully digital bound is very closed, with a penalty lower than the penalty of ULA case. This occurs because the proposed equalizer explores the correlative feature of channel, and thus, with the UPA configuration where the correlative level of channel is higher than ULA case, the proposed equalizer is more efficient.

Finally, in Figure 11 and Figure 12, we evaluate the impact of the path loss and the shadowing effects on the performance. As it can be seen the proposed algorithm outperforms the two-step one, with just four iterations. Also, the proposed algorithm almost achieves the fully digital bound. In general, we can draw the same conclusions as those drawn for scenarios without path loss and showing effects.

## 6. Conclusions

In this paper, we developed a fully iterative analog–digital multi-user equalizer approach for wideband mmWave mMIMO SC-FDMA systems. In the proposed approach the digital and analog parts are computed to efficiently remove the residual multi-user and inter-symbol interferences. The equalizer was designed by using the sum of the MSE as the minimization metric. In the design it was assumed that the analog part is constant over the subcarriers because of hardware constraints.

The results showed that the proposed multi-user equalizer is very efficient to remove the multi-user/inter-symbol interference, achieving a performance close to the digital counterpart for the both antenna array configurations, ULA and UPA, and for scenarios with and without path loss and shadowing effects. Furthermore, the proposed iterative approach outperforms the two-step one previously proposed, at the cost of some more complexity. Therefore, the proposed iterative hybrid multi-user equalizer could be an interesting approach for practical scenarios that requires high reliability links.

## Figures and Tables

**Figure 1 sensors-20-00575-f001:**
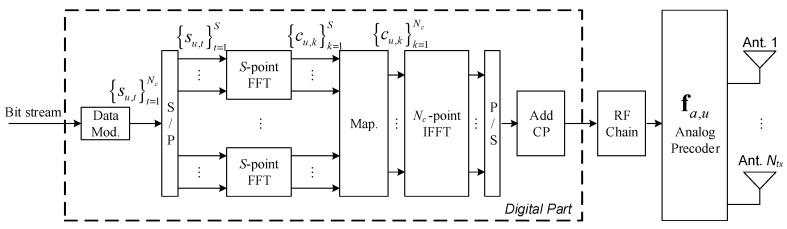
General block diagram of the *u*th user terminal.

**Figure 2 sensors-20-00575-f002:**
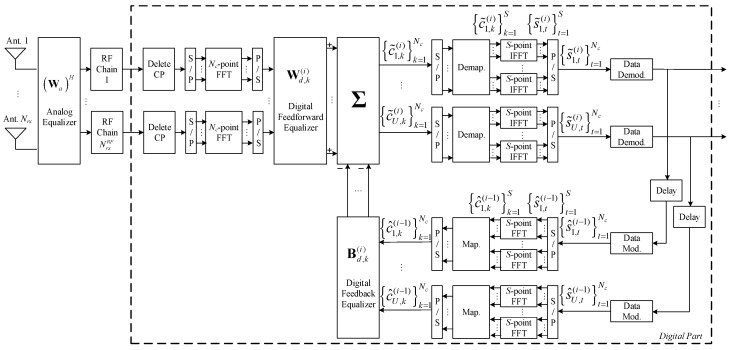
Proposed receiver structure.

**Figure 3 sensors-20-00575-f003:**
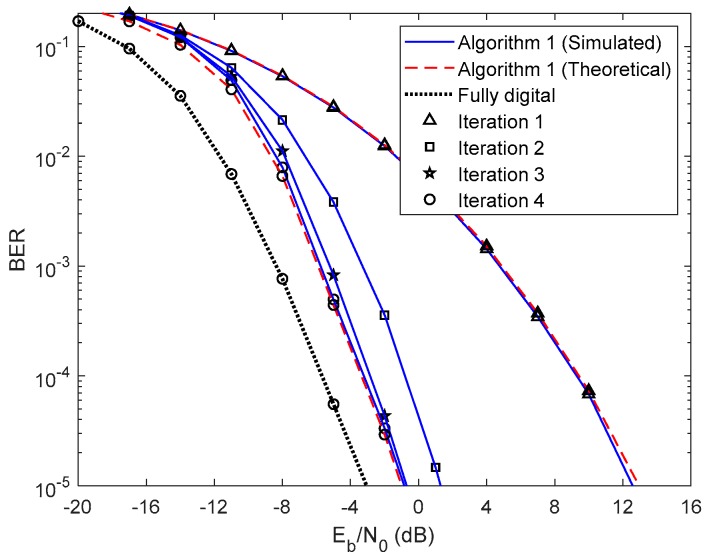
Performance of the proposed hybrid iterative multi-user equalizer (Algorithm 1) for the NCSI precoder with ULA configuration and QPSK modulation.

**Figure 4 sensors-20-00575-f004:**
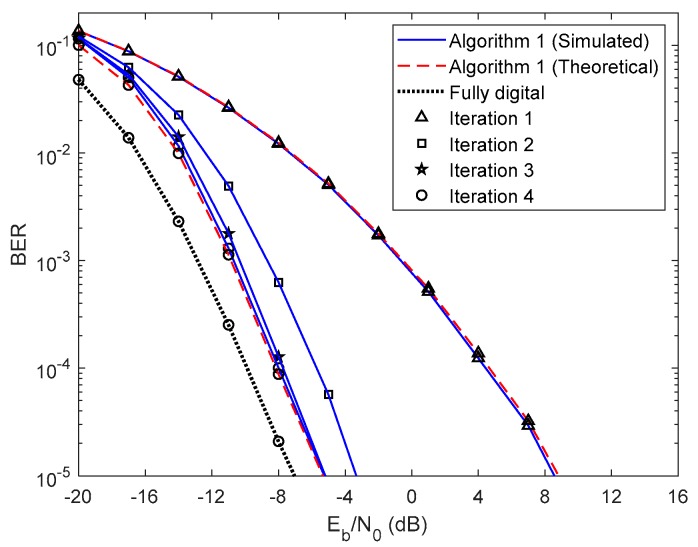
Performance of the proposed hybrid iterative multi-user equalizer (Algorithm 1) for the PCSI precoder with ULA configuration and QPSK modulation.

**Figure 5 sensors-20-00575-f005:**
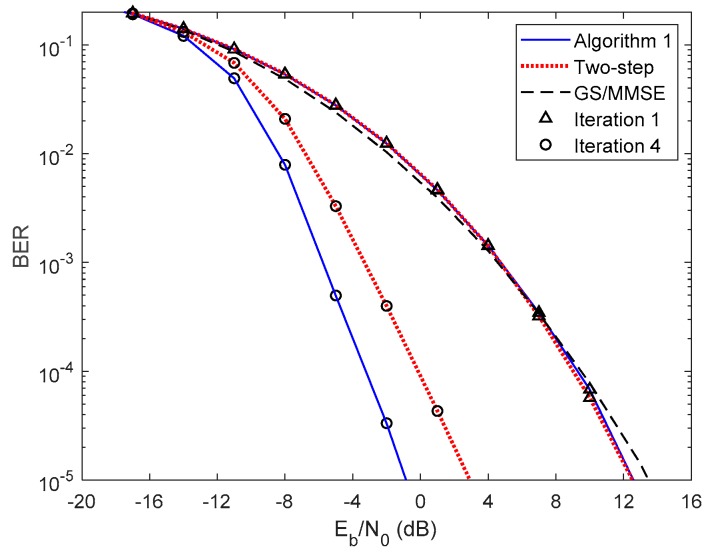
Performance comparison between the Algorithm 1, the two-step, and GS/MMSE approaches for the NCSI precoder with ULA configuration and QPSK modulation.

**Figure 6 sensors-20-00575-f006:**
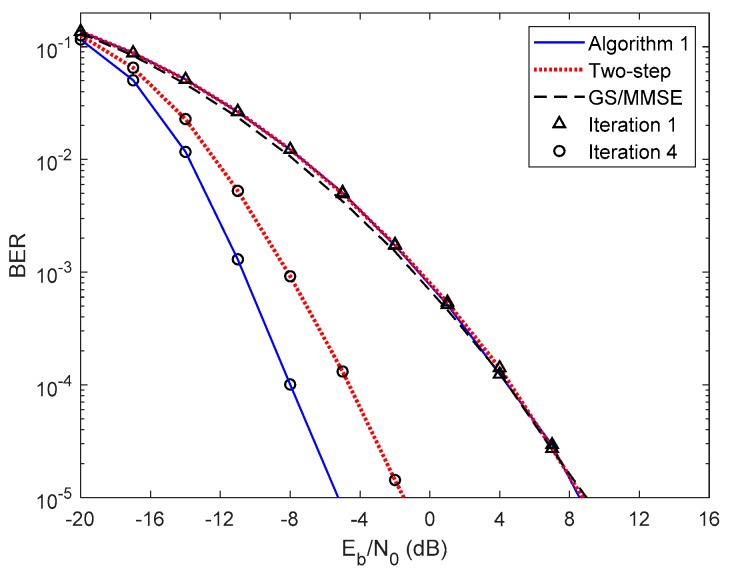
Performance comparison between the Algorithm 1, the two-step, and GS/MMSE approaches for the PCSI precoder with ULA configuration and QPSK modulation.

**Figure 7 sensors-20-00575-f007:**
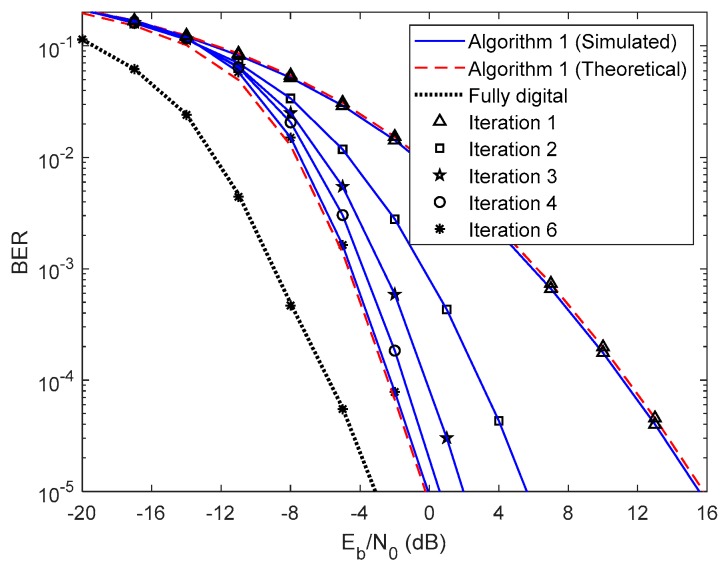
Performance of the proposed hybrid iterative multi-user equalizer (Algorithm 1) for the PCSI precoder with ULA configuration and 16 QAM modulation.

**Figure 8 sensors-20-00575-f008:**
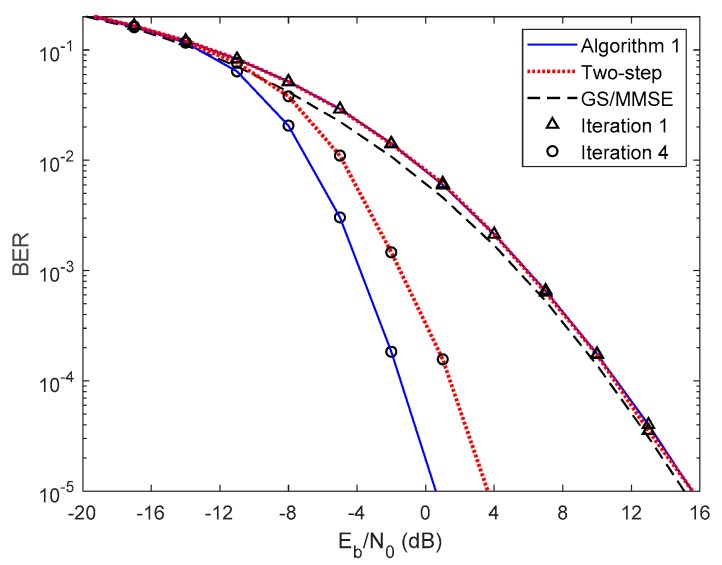
Performance comparison between the Algorithm 1, the two-step and GS/MMSE approaches for the PCSI precoder with ULA configuration and 16 QAM modulation.

**Figure 9 sensors-20-00575-f009:**
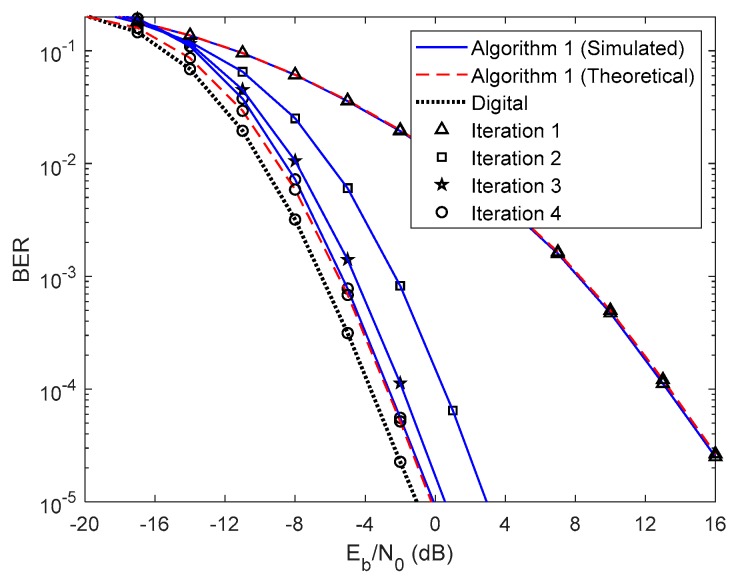
Performance of the proposed hybrid iterative multi-user equalizer (Algorithm 1) for the NCSI precoder with UPA configuration and QPSK modulation.

**Figure 10 sensors-20-00575-f010:**
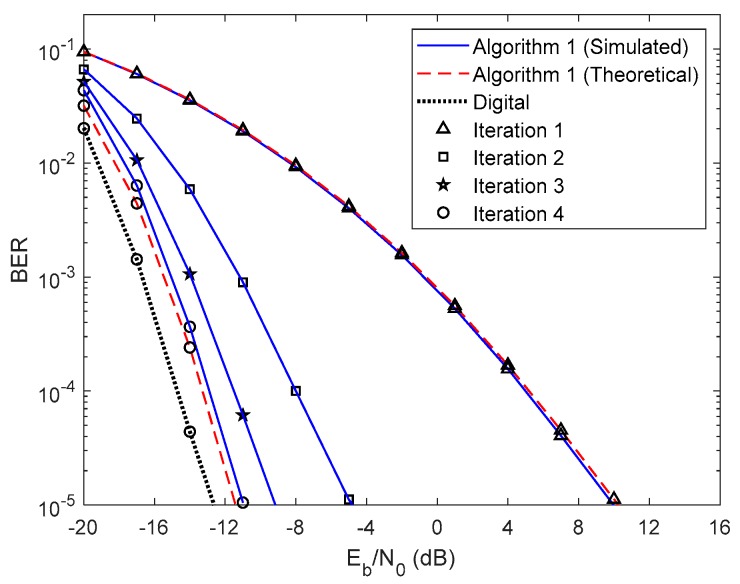
Performance of the proposed hybrid iterative multi-user equalizer (Algorithm 1) for the PCSI precoder with UPA configuration and QPSK modulation.

**Figure 11 sensors-20-00575-f011:**
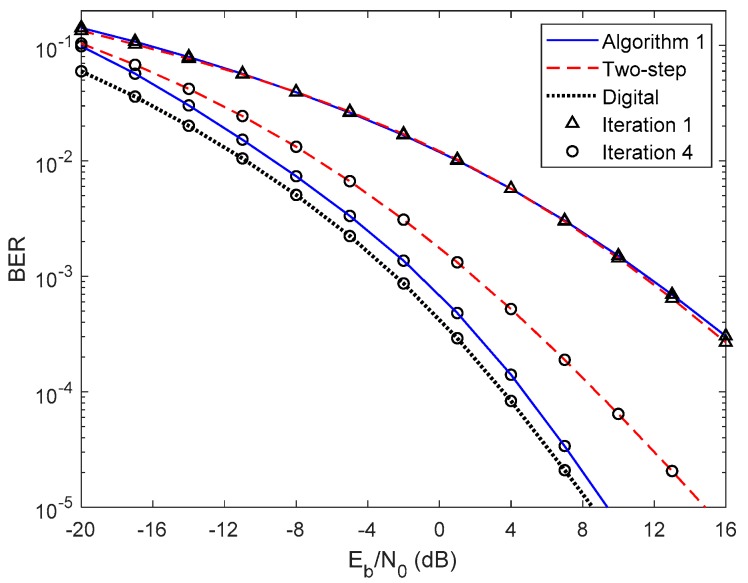
Performance comparison between the Algorithm 1 and the two-step approach for the NCSI precoder with ULA configuration, path loss, and shadowing and QPSK modulation.

**Figure 12 sensors-20-00575-f012:**
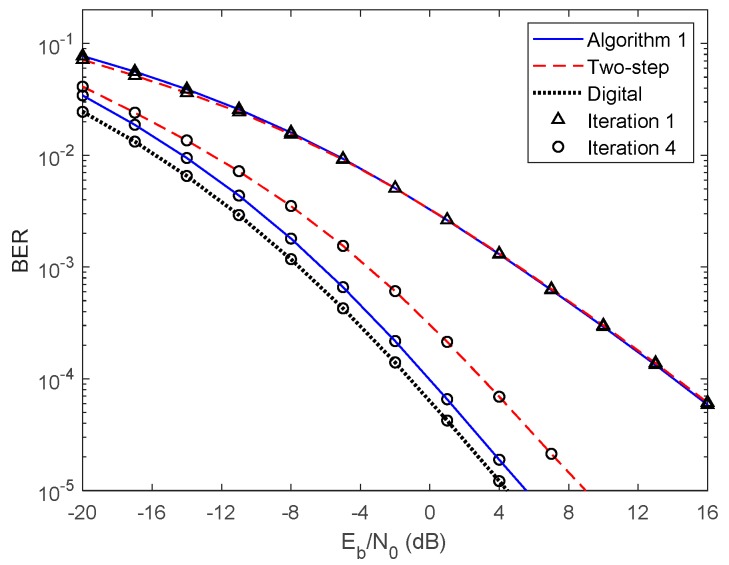
Performance comparison between the Algorithm 1 and the two-step approach for the PCSI precoder with ULA configuration, path loss, and shadowing and QPSK modulation.

**Table 1 sensors-20-00575-t001:** Simulation parameters.

Parameter	Value
Carrier frequency	72 GHz
Antenna element spacing	Half-wavelength
Array configuration	ULA or UPA
$N_{cl}$	4
$N_{ray}$	5
$\sigma_{\phi_{q}^{rx}}^{},\sigma_{\phi_{q}^{tx}}^{}$	10 degrees
$N_{c}$	512
D	128
S	128
U	8
$N_{tx}$	16
$N_{rx}$	32
$N_{rx}^{RF}$	8
Modulation	QPSK or 16-QAM

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
