# Peer review of "Iterative Analog–Digital Multi-User Equalizer for Wideband Millimeter Wave Massive MIMO Systems"

_sensors, 2020, doi:10.3390/s20020575_

Round 1

Reviewer 1 Report

The Reviewer appreciates the work of revision by the authors. However, my previous comments on novelty and quality of presentation remain unchanged.

Novelty. Despite the stated differences in hardware, it is still a fact that the proposed problem and the one from [24], now [29] are formally almost the same. The minor differences highlighted by the authors only result in an additional step of optimization, as the almost equivalent problem is still addressed by alternating between two sub-problems. The proposed approach is known to be sub-optimal and makes the contribution of the proposed work still rather limited.

Quality of written English still require improvements. Many sentences have verbs separated by subjects, many sentences are still awkward. Some of the ones highlighted by the Reviewer in the previous round of revisions have basically remained unchanged.

Reviewer 2 Report

This paper proposes an iterative analog-digital multi-user equalizer scheme for the uplink of wide-band millimeter-wave massive multiple-input-multiple-output (MIMO) systems. From a reviewer point, the mathematical calculations seem to be correct. However, I have the following questions:

(1)The key challenges of the work are not explained clearly.

(2)How the transmitter know the partial CSI?

(3)In this paper, the authors should compare the proposed method with more new methods .This is very important.

(4)Is the complexity of the proposed algorithm very high? Is it difficult to use in practice?The authors also should compare the complexity of the proposed method with other schemes.

(5)It is generally known that some algorithms are sensitive to the initial values. I want to know whether the initial value of the proposed algorithm can influence the results or not? And how do you select the initial value?

(6)In the simulation, the detailed parameters should be listed in the table.

(7)The discussion of MIMO related works is not enough. For example, the following need to be discussed:“GWO-BP neural network based OP performance prediction for mobile multiuser communication networks,” IEEE Access, 2019,7:152690-152700; “Outage Performance for IDF Relaying Mobile Cooperative Networks”, Mobile Networks & Applications,2018,23 (12) :1496-1501.

Reviewer 3 Report

This paper proposes an iterative method of uplink equalizer for wideband mmWave massive MIMO systems. There are three main design targets: 1) analog precoding matrix, 2) digital precoding matrix, and 3) digital feedback matrix. By defining the MSE as (13), it formulates an optimization problem to minimize the sum MSE (16). In an iterative fashion, it obtains the optimal matrix by assuming that the other two are determined. 

Generally, I believe this paper studies an adequate problem and the proposed scheme is novel enough. Before publication, however, I suggest the authors go through another round of revision to improve the paper's quality. Here is my comment.

1) In (11), the channel equalization is performed by W_d, W_a, and B, and B (feedback matrix) is useful only if we use SC-FDMA. This is because there is no ISI by using OFDM, so that only inter-user interference matters. What I wonder is that how much performance gains are obtained by using the feedback matrix B. I suggest add the performance comparison by setting B = 0.

2) I need more explanation in the equation (12). What is the intuition to express (12)?

3) For more clear understanding, it is better to include the detail MSE derivation (14).

4) I guess finding the analog precoder by using (30) needs exhaustive search. If this is true then why do we need (30) since we need to search over all the feasible codebook indices anyway.

5) The performance comparison will become more convincing if the baseline methods are added.

Round 2

Reviewer 2 Report

It is well revised.

Reviewer 3 Report

The authors addressed my comments well. I have no further concern.

This manuscript is a resubmission of an earlier submission. The following is a list of the peer review reports and author responses from that submission.

Round 1

Reviewer 1 Report

The paper describes an iterative analog-digital equalizer for mmwave communication systems. The topic is timey and interesting, however the reviewer has the following concerns:

Novelty. The paper draws significantly and extensively from previous works of the authors and contributions in the proposed manuscript are very straightforward and trivial. Analytical derivations follow nearly step by step previous work in [24]. It is not even clear how the two works differ, as in the paper the authors state that the joint optimization problem (21) is equivalent to simply (27), which is the optimization for the analog component alone. The analog component is then used to perform the optimization on the digital component. It follows that no joint optimization is performed and only an alternate optimization between two separate problems is performed. The inclusion of pseudocode would have helped improve understanding, however, despite being declared in the text, it is missing. Quality of presentation. Quality of presentation is rather poor. As mentioned in the previous sections, the authors refer to Algorithm 1, yet do not include it in the text. Multiple typos and improper wording can be found throughout the manuscript: "we assume ate the receiver", "which it can be shown", "is compared with other recently considered approach", "the receive power" and so on. Additionally, many sentences are imprecise and not suitable for a journal scientific publication: "there is an overcrowding of conventional bands", "are jointly optimized using the sum of the mean square error over the subcarriers as metric", the complexity of the proposed fully iterative analog-digital is higher than our previous two-step approached but the performance is better" and "are computed jointly to remove interferences". The list is just a handful example of significant more instances throughout the manuscript.